# Comparison of six methods for *Loa loa* genomic DNA extraction

**Roland Dieki[1,2], Elsa-Rush Eyang-Assengone[3], Patrice Makouloutou-Nzassi[3], Félicien Bangueboussa[3], Edouard Nsi Emvo[2†], Jean Paul Akue [1]\***

**1** DÃpartement de Parasitologie, Centre International de Recherches MÃdicales de Franceville (CIRMF), Franceville, Gabon, **2** Laboratoire de Recherche en Biochimie (LAREBIO), FacultÃ des Sciences, UniversitÃ des Sciences et Techniques de Masuku (USTM), Franceville, Gabon, **3** Unité de Recherches en Ecologie de la Santé (URES/CIRMF), BP. 769 Franceville, Gabon

† Deceased.
\* jpakue@yahoo.fr

**Data Availability Statement:** All relevant data are within the paper and its Supporting information files.

## Abstract

### Objectives

Good-quality and sufficient DNA is essential for diagnostics and vaccine development. We aimed to compare six DNA extraction techniques applied to *Loa loa* microfilariae in order to evaluate the purity and integrity of extracts in terms of quality and quantity.

### Methods

The microfilariae were purified via a Percoll gradient procedure with blood from hyper-microfilaremic individuals (> 30,000 microfilaria [mf]/ml). DNA extraction was carried out in duplicate at a rate of 350,000 mf/tube for each technique: phenol/chloroform, commercial Qiagen kit, salting out, Tris-EDTA, methanol, and cetyltrimethylammonium bromide (CTAB). The integrity, purity, concentration, and quality of the DNA extracts were successively verified by agarose gel electrophoresis, spectrophotometry ($A_{260}/A_{280}$ and $A_{260}/A_{230}$ wavelength ratio), Qubit fluorometry, and endonuclease and polymerase activity. The six techniques were compared on the basis of the following parameters: concentration, purity, efficiency, effectiveness, integrity, safety of the technique, as well as cost and duration of the protocol.

### Results

The ratios of the optical densities of the extracts $A_{260}/A_{280}$ and $A_{260}/A_{230}$ were, respectively: phenol/chloroform (1.82; 1.11), Qiagen (1.93; 1.36), salting-out (1.9; 2.04), Tris-EDTA (1.99; 1.183), methanol (2.126; 1.343), and CTAB (2.01; 2.426). The DNA yield was: phenol/chloroform (3.920 μg), Qiagen (10.280 μg), salting-out (10.390 μg), Tris-EDTA (0.5528 μg), methanol (0.1036 μg), and CTAB (1.115 μg). Endonuclease and polymerase activity was demonstrated by digestion of DNA and through amplicons obtained

**Funding:** The authors received no specific funding for this work.

**Competing interests:** The authors have declared that no competing interestq exist.

via polymerase chain reaction assays with phenol/chloroform, Qiagen, and salting-out extracts.

## Conclusion

The phenol/chloroform, Qiagen, and salting-out DNA extracts were all of good quality. Salting out had the best yield followed by Qiagen and then phenol/chloroform. Endonuclease and polymerase activity was effective in all three extracts despite the presence of some contaminants. These methods are therefore suitable for the extraction of DNA from *Loa loa* microfilariae. Tris-EDTA and methanol did not show adequate sensitivity, while CTAB was found to be unsuitable.

## Introduction

*Loa loa* has received increasing attention during the past three decades because of its role in hindering the implementation of the World Health Organization's mass drug administration strategy in regions with co-endemicity of *L. loa* and other filarial parasites [1]. The clinical presentation of loiasis is complex and makes the diagnosis challenging; moreover, 70% of infected individuals do not have circulating microfilariae in the peripheral blood, and the only point-of-care test available is the detection of microfilariae in the blood or passage of an adult worm under the conjunctiva (eye worm) [2–4]. Although these methods are specific, they are not sensitive in detecting all cases of loiasis. Furthermore, any strategy to control loiasis will require methods that can evaluate the success or failure of the strategy. Therefore, good diagnostic methods are needed for *L. loa*. The observation that 70% of infected individuals are amicrofilaremic suggests the existence of active immunity against the larval stage of this filarial parasite. This finding calls for the development of a vaccine strategy to control the spread of filarial worms. For the development both of diagnostic methods and of vaccines, nucleic acid is a crucial basic material; first, because many diagnostic methods based on DNA detection have shown good specificity and sensitivity [5, 6]. Second, the study of *L. loa* has been hampered by the lack of an appropriate animal model that can provide sufficient material of the parasite stage. Thus, high-purity DNA is required for diagnostic tools or for cloning of important genes that can provide recombinant DNA molecules necessary for the evaluation of vaccines or chemotherapy targets. Extraction of genomic DNA of *L. loa* has been achieved using adult worms for molecular cloning [7] and to search for repeated DNA sequences for specific detection of *L. loa* in vectors [8]. The classic phenol/chloroform method has been used for DNA extraction. In both cases, the number of adult worms extracted from humans was limited. Many techniques used in this context require radioactive isotopes, which are associated with increased hazard for the operator [9]. Another source of accessible DNA is microfilariae in the blood [10]. Whether for diagnostic or for cloning purposes, extraction of DNA with the phenol/chloroform method is time-consuming and can be hazardous. Several methods have been developed and are in commercial DNA extraction kits, while other methods that exist have never been applied to *L. loa*, for example, those using cetyltrimethylammonium bromide (CTAB), Tris-EDTA, methanol, and salting-out. These methods may offer advantages in terms of cost, time, quality, quantity, and hazard exposure. To date, no study has been conducted to compare the efficiency of these techniques for the extraction of *L. loa* DNA. In this study, we evaluated six techniques for extraction of *L. loa* DNA, namely, methods using phenol/chloroform, Qiagen, salting-out, Tris-EDTA, methanol, and CTAB.

## Material and methods

### Isolation and purification of microfilariae

The sampling was carried out in a rural area in the village Olounga, located 74 km from Franceville in the department of Sébé-Brikolo, province of Haut-Ogooué, in south-east Gabon. Individuals with high loads of microfilariae (30,000 microfilariae per mL of peripheral blood) were selected for the study after informed consent signed in accordance with the protocol approved by The National Ethics Committee of Gabon (PROT N00001/20/6/3/SG/CNE). A total of 500 mL of blood was drawn by venous puncture from the arm of infected individuals and microfilariae were isolated and purified as follows [11]:

Iso-osmotic Percoll stock (SIP) was prepared by adding 9 volumes of Percoll and 1 volume of RPMI-1640 medium (10×). Gradient solutions containing 40%, 50%, and 65% SIP were prepared and diluted with RPMI-1640 medium (1×). In 15-ml polystyrene tubes and using Pasteur pipettes, the gradients were carefully added in turn, 4 ml of the 65% gradient, 2 ml of the 50% gradient, and 2 ml of the 40% gradient. Over the gradient layers, 2 ml of whole blood from infected individuals was added. The tubes were centrifuged at 1500 rpm for 20 min. After centrifugation, different layers appeared. The layers containing the microfilariae were introduced into a device consisting of a syringe and a filter with 5-μm pores. The solution was passed through the filter by applying pressure on the syringe. The device was then disassembled and the filters were incubated for 5 min in Petri dishes containing RPMI-1640 medium (1×) to let the microfilariae out. The microfilariae were concentrated by centrifugation and were stored at -20°C.

### Distribution of the microfilariae

A total of 12 tubes, i.e., two tubes per method, containing 350,000 microfilariae per tube were used. Six DNA extraction protocols were performed: phenol/chloroform, Qiagen, salting-out, Tris-EDTA, methanol, and CTAB extraction.

### DNA extraction using phenol/chloroform

The lysis buffer was composed of 50 mM Tris-HCl pH 8, 100 mM NaCl, 50 mM EDTA, 1% SDS, 30 mM β-mercaptoethanol, 1500 μl/ml (10mg/ml stock) proteinase K, and water.

The extraction was done according to Barker's procedure [12]. The microfilariae were incubated with 500 μl of lysis buffer for 1 h 30 min at 65°C in a water bath. The supernatant was then recovered. The DNA from the microfilariae recovered in the supernatant was extracted with 250 μl/250 μl of phenol/chloroform three times. The three extractions were followed by centrifugation at 14,000 rpm for 3 min. The aqueous phase was recovered each time. The DNA was precipitated in 2 volumes of 95% ethanol in the presence of 0.1 volume of sodium acetate. The tubes were subsequently incubated for 20 min at -20°C, after which they were centrifuged at 14,000 rpm for 30 min. The extracted DNA was washed three times in succession with 250 μl of 70% cold ethanol. The extracted DNA was stored in water solution at -20°C.

### DNA extraction using the Qiagen kit

The extraction was performed following the manufacturer's instructions (DNeasy® Blood & Tissue Kit). In a 1.5-ml microcentrifuge tube containing the isolated and purified microfilariae, 180 μl of the ATL buffer solution and 20 μl proteinase K were added; the sample was then incubated at 56°C in a water bath for 1 h. Subsequently, 200 μl of the buffer AL solution was added to the tubes. The tubes were then incubated again in a water bath at 56°C for 10 min. After this second incubation, 200 μl of 100% ethanol was added. Using a pipette, each mixture

was transferred into mini-chromatography tubes. The mixtures were then centrifuged for 1 min at 8000 rpm. After centrifugation, the DNA adsorbed on the silica membrane was washed twice: first with 500 µl of AW1 buffer solution followed by centrifugation for 1 min at 14,000 rpm, and then with 500 µl of AW2 buffer solution with centrifugation at 14,000 rpm for 3 min. A total of 200 µl of elution solution AE was introduced at the center of the mini-chromatography columns and then incubated for 1 min at room temperature. This step was followed by centrifugation for 1 min at 8000 rpm. The DNA extracted was stored in elution buffer provided by the manufacturer at -20°C.

## DNA extraction using salting-out

The buffer for salting-out was composed of 0.4 M NaCl, 10 mM Tris-HCl pH 8, and 2 mM EDTA. The extraction protocol was standardized in 1988 by Miller et al. [13]. Initially, 400 µl of buffer was added to the tubes containing the isolated and purified microfilariae. The resulting solution was homogenized with ultrasound for 10–15 s. Then, 40 µl of 20% SDS and 8 µl of 20 mg/ml proteinase K were added to the tubes and mixed. The samples were incubated at 65°C in a water bath for 1 h. After incubation, 300 µl of 6 M $NaCl_2$ was added to each tube, mixed, and then centrifuged for 30 min at 10,000 $g$. The resulting supernatant was transferred to another tube and an equal volume of isopropanol was added. The tubes were incubated at -20°C for 1 h and then centrifuged for 20 min at 40°C and 10,000 $g$. The supernatant was discarded and the pellet was washed by centrifugation with 70° ethanol for 20 min. The pellet was then dried and suspended in 200 µl of water. The DNA obtained was stored at -20°C.

## DNA extraction using Tris-EDTA (TE)

The buffer consisted of 10 mM Tris base, Tris-HCl pH 8, and 0.1 mM EDTA.

In the tubes containing the isolated microfilariae, 65 µl of TE was added and incubated for 15 min at 50°C in a water bath. The tubes were transferred to another water bath and incubated at 97°C for 15 min. The tubes were then centrifuged for 1 min and the resulting DNA was stored at -20°C.

## DNA extraction using methanol

In tubes containing the microfilariae, 125 µl of methanol was added and incubated at room temperature for 15 min. The methanol was removed by centrifugation for 5 min at 15,000 rpm and dried. After drying the tubes, 65 µl of distilled water was added to each tube and incubated in a water bath at 97°C for 15 min. The DNA extracted was kept at -20°C.

## DNA extraction using CTAB

The CTAB buffer was composed of 20 g/l CTAB, 1.4 M NaCl, 0.1 M Tris-HCl, 20 mM $Na_2EDTA$, and water. The pH was adjusted to 8.0 by adding 1 M NaOH. The CTAB precipitation solution was composed of 5 g/l CTAB, 0.04 M NaCl, and water.

In sterile 1.5-ml tubes containing the isolated and purified microfilariae, 300 µl of sterile deionized water was added. Subsequently, 500 µl of CTAB buffer and 20 µl of proteinase K (20 mg/ml) were added to the tubes. The whole sample was mixed and incubated at 65°C for 90 min. This was followed by adding 20 µl RNAse A (10mg/ml) to the tubes, mixing, and incubating the tubes at 65°C for 10 min. The tubes were then centrifuged for 10 min at 16,000 $g$. The supernatant liquids were transferred to microcentrifuge tubes containing 500 µl of chloroform and the tubes were mixed for 30 s. The tubes were subsequently centrifuged for 10 min at 16,000 $g$. The upper layers were transferred to new microcentrifuge tubes, and 2 volumes of

CTAB precipitation solution was added and mixed by pipetting. The tubes were incubated for 60 min at room temperature and centrifuged for 5 min at 16,000 *g*.

The supernatant liquid was discarded. In the precipitate obtained, 350 μl of 1.2 M NaCl was dissolved and 350 μl of chloroform was added and mixed for 30 s. Everything was centrifuged for 10 min at 16,000 *g* until phase separation occurred. The upper layer was transferred to a new microcentrifuge tube and 0.6 volume of isopropanol was added. The tubes were centrifuged for 10 min at 16,000 *g*. Subsequently, 500 μl of 70% ethanol was added to the collected supernatant. The tubes were centrifuged for 10 min at 16,000 *g*. After centrifugation, the supernatant liquids were discarded and the pellets containing the DNA were dissolved in 100 μl of sterile deionized water. The DNA solution was stored at -20˚C.

## Quantification and qualitative analysis of extracted DNA

The DNA concentrations were measured by the Qubit® 2.0 method (S2 Table). Extracted DNA was analyzed spectrophotometrically using NanoVue™, with different optical densities (OD) at 230-, 260-, 280-, and 320-nm wavelengths measured for each sample. The purity of DNA was assessed based on calculation of the 260/280 and 260/230 ratios after subtraction of each OD obtained at 230, 260, and 280 nm with the OD at 320 nm of the same sample (S2 Table). The interpretations of these analyses are based on the well-known classic rule [14] described in the supporting information (S1 Table).

## Endonuclease activity

In order to evaluate the endonuclease activity, EcoRI and BamHI endonucleases were used following the protocol provided by the manufacturer. Briefly, 10 μL of DNA extracted by each method was mixed with 10× buffer, EcoRI (20,000 U/ml) and BamHI (100,000 U/ml) enzymes and with water up to 50 μl. Digestion was performed at 37˚C for 2 h and stopped at 65˚C for 10 min. The results were analyzed by electrophoresis on a 1% agarose gel.

## Polymerase activity

PCR was performed by using primers designed from the ALT-1 gene of *Brugia malayi* L3 larvae. The reaction was carried out by mixing 2 μl of 10 mM solution of a mixture of the four dNTPs (dATP, dCTP, dGTP, and dTTP at 10 mM each), 10 μM of each primer (ALT-1 F and ALT-1 R), 10 μl of PCR buffer, 10 μl of DNA extract, and 2.5 U of Taq polymerase. Sterile water was added to achieve a volume of 100 μl. The reaction tubes were placed in a thermal cycler (PerkinElmer GeneAmp PCR System 2400) for 35 cycles in the following steps: 95˚C for 5 min, 94˚C for 1 min for melting, 1 min at 45˚C for annealing, 2 min at 72˚C for extension, and with a final extension at 72˚C for 5 min. Owing to the similarities between *Brugia* and *L. loa* [15], we used primers derived from abundant larval transcript-1 from *Brugia malayi* [16]. The primers were 5′GAT-GAC-GAA-TTC-GAC-GAC-GAA-TCC-TCA3′ and 5′TTG-TTT-TGC-TTG-CTT-TGT-AAG-CAT-TTA3′. The PCR products were analyzed on 1.5% agarose gels.

## Agarose gel electrophoresis

A total of 10 μl of sample diluted in sample buffer was loaded in a well of 0.8% (genomic DNA extract), 1% (enzyme restriction for DNA), and 1.5% (PCR amplicon) agarose gel. Migration was performed in TBE buffer under 120 V for 1 h gels. DNA was revealed by GelRed staining.

**Table 1. Comparative analysis of spectrometric data from six DNA extraction methods.**

| Method | No. of microfilariae/tube[*] | Purity (NanoVue)[a,b] | | Quantity (Qubit)[c,d] | | Hazard | Processing time | Cost per sample |
|---|---|---|---|---|---|---|---|---|
| | | Ratio 260/280 | Ratio 260/230 | C˚ (ng/ul) | Yield (µg) | | | |
| Phenol/chloroform | 350,000 | 1.82 | 1.11 | 39.2 | 3.92 | Yes | 2 h 29 min | Expensive |
| Qiagen | 350,000 | 1.93 | 1.36 | 51.4 | 10.28 | No | 1 h 21 min | Very expensive |
| Salting-out | 350,000 | 1.9 | 2.04 | 51.95 | 10.39 | No | 2 h 17 min | Cheap |
| Tris-EDTA | 350,000 | 1.994 | 1.183 | 8.505 | 0.5528 | No | 32 min | Cheap |
| Methanol | 350,000 | 2.126 | 1.343 | 1.595 | 0.1036 | No | 38 min | Cheap |
| CTAB | 350,000 | 2.01 | 2.426 | 11.15 | 1.115 | Yes | 4 h 8 min | Expensive |

[*]: In each tube X 2 experiments per method;

[a]: Using the mean OD of 2 experiment for each method S2 Table;

[b]: Using mean OD of 2 experiments for each method S2 Table;

[c]: Mean value of 2 experiment for each method S2 Table;

[d]: Yield according to the formula in S1 Table.

## Results

### Comparative spectrometric analysis of the six methods

Measurement of OD at 230, 260, 280, and 320 nm (Table 1) revealed an almost similar 260/ 280 ratio between phenol/chloroform, Qiagen, salting-out, and Tris-EDTA (1.82, 1.93, 1.9, 1.994, respectively), whereas methanol and CTAB had the highest ratio (2.126 and 2.01 respectively). When the 260/230 ratio was compared, it was lower with phenol/chloroform and Tris-EDTA (1.11 and 1.183, respectively) followed by methanol and Qiagen (1.343 and 1.36, respectively). The highest 260/230 ratio was obtained with salting-out and CTAB (2.04 and 2.426, respectively). In the Qubit analysis (Table 1), the concentration of DNA was higher with salting-out (51.9 ng/µl) followed by Qiagen (51.4 ng/µl) and phenol/chloroform (39.2 ng/µl). These concentrations were far greater than CTAB, Tris-EDTA, and methanol (11.15 ng/µl, 8.05 ng/µl and 1.595 ng/µl, respectively). As a consequence (Table 1), the yield per method was 3.920 µg, 10.280 µg, 10.390 µg, 0.5528 µg, 0.1036 µg and 1.115 µg for phenol/chloroform, Qiagen, salting-out, Tris-EDTA, methanol, and CTAB, respectively, from a sample stock of 350,000 *Loa loa* microfilariae. At the end of this process, we evaluated each technique according to the criteria listed in S1 Table. Phenol/chloroform and CTAB involved some hazard. The time required to perform the CTAB method is the longest (4 h 6 min) followed by phenol/ chloroform, salting-out, and Qiagen. The Qiagen method was the costliest, followed by phenol/chloroform and CTAB, while the other three methods were comparatively cheap.

### Gel electrophoresis analysis

When the DNA extracted by the phenol/chloroform technique was analyzed on a 0.8% agarose gel, a high-molecular-weight band above the standard molecular weight appeared followed by a smear (Fig 1); similar bands were seen with Qiagen and salting-out. However, these bands were almost absent with the Tris-EDTA, methanol, and CTAB methods, which presented a smear within the range of molecular-weight markers (Fig 1, lines 7–12). Moreover, the salting-out method showed a very strong slurry aspect around the main band (Fig 1, lines 5–6).

### Enzyme activity of DNA

In order to evaluate the integrity of the extracted DNA and to ensure that the extract was identical notwithstanding the different methods used, two families of enzymes were tested:

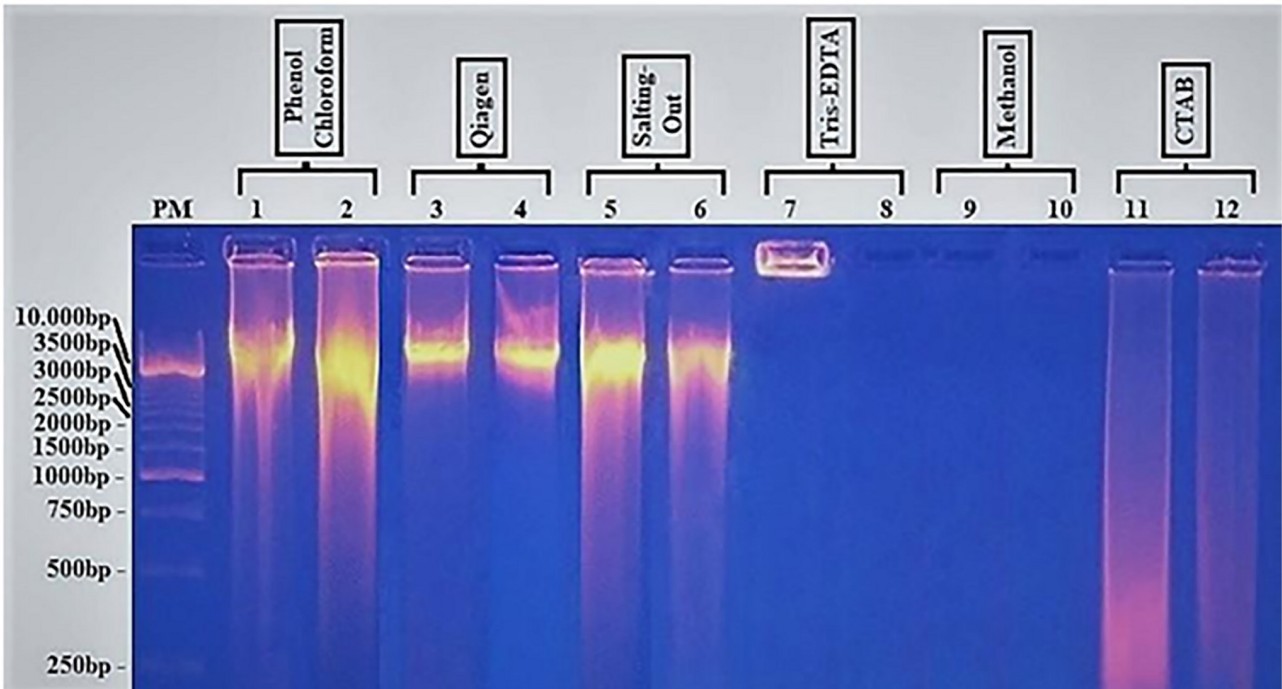

**Fig 1. Comparison of six methods of *Loa loa* DNA extraction.** Extracted DNA in duplicate was analyzed by electrophoresis in a 0.8% agarose gel and visualized under UV light. PM = standard molecular weight. Lines 1–2: DNA from phenol/chloroform; 3–4: Qiagen extract; 5–6: Salting-out extract; 7–8: Tris-EDTA extract; 9–10: Methanol extract; 11–12: CTAB extract. The values on the left are the size of molecules.

endonuclease class II and polymerase. For endonuclease, two enzymes were used. EcoRI digestion of all DNA extracted by all six methods showed a shift in molecular size of DNA from high to low molecular weight for DNA extracted by phenol/chloroform, Qiagen, and salting-out; however, this was not distinguishable for the rest of the methods (Fig 2A). The shift was identical in size regardless of the technique used. This observation was also made with the BamHI enzyme (Fig 2B).

For polymerase activity, a PCR test using Taq polymerase enzyme was used for amplification of the *ALT1* gene. The results showed amplicons for phenol/chloroform, Qiagen, and salting-out extracts with a size of 700 bp (Fig 3, lines 1–3). However, for the other methods, no amplicon was visible (Fig 3, lines 4–5). Only a smear was seen for CTAB extract (line 6, Fig 3).

## Discussion

The quality parameter of the genomic DNA extracted is an essential criterion for optimizing the sensitivity and specificity of molecular biology and genetic engineering techniques. Although the exact nature of the DNA contaminants was unknown, we based our hypothesis or suggestions on criteria defined in the supporting information (S1 Table) and substantiated by commonly used classic parameters described by Surzycki [14].

Several techniques for DNA extraction are used in field work and in endemic countries with poor settings. It is important to know the advantages and disadvantages of these techniques with regard to the safety, cost, yield, and–most importantly–the quality of DNA. However, such analyses have not been conducted in the field of *L. loa* research. In this study, we examined six methods for DNA extraction by comparing classic approaches (phenol/chloroform and CTAB), simple methods (Tris-EDTA, methanol, and salting-out), and commercial

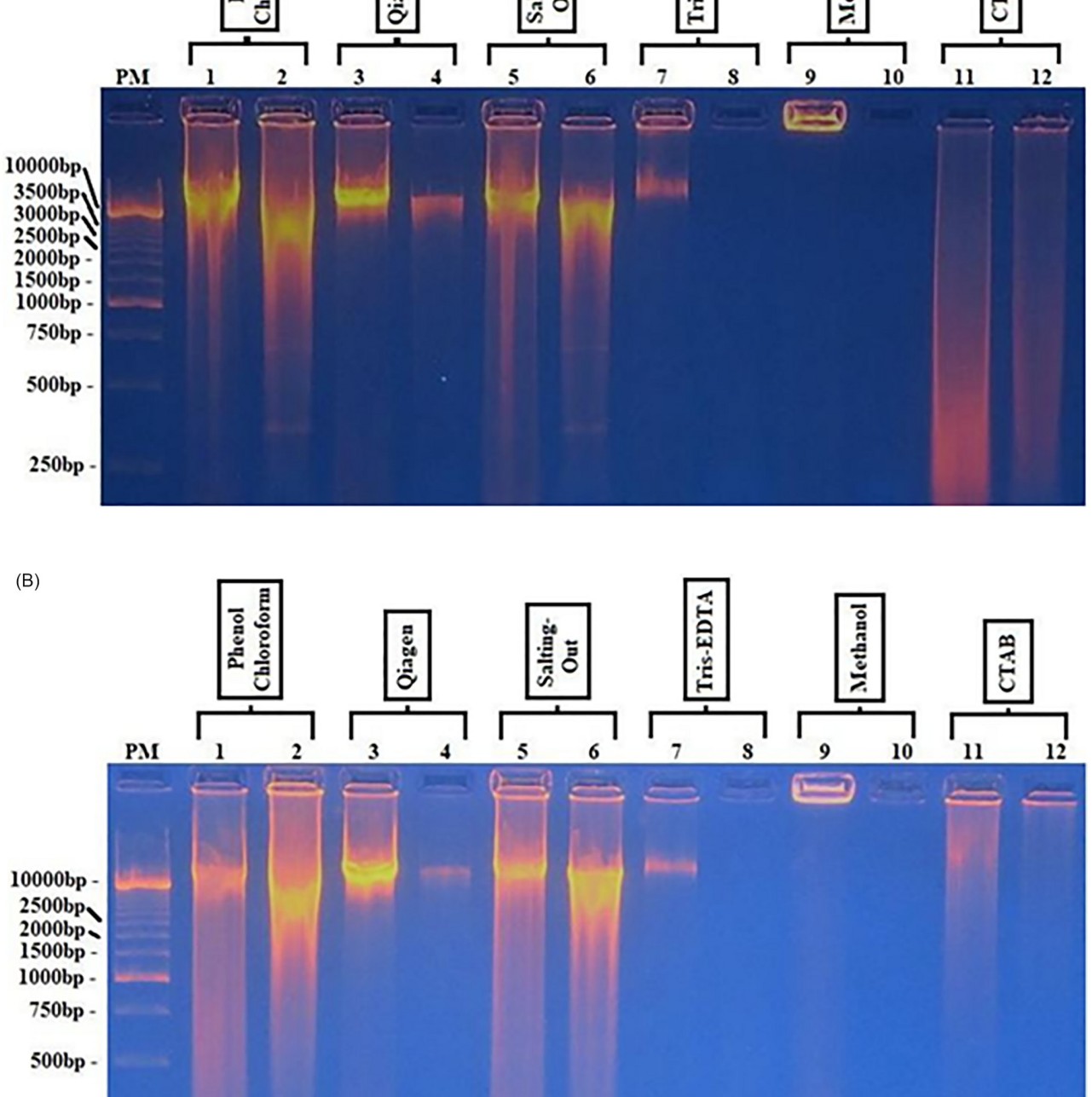

**Fig 2. Digestion of extracted *Loa loa* DNA by EcoRI and BamHI endonuclease. 2A**: Undigested (lines: 1-3-5-7-9-11) compared to digested with **EcoRI** (lines: 2-4-6-8-10-12) DNA; extracts were analyzed via agarose gel electrophoresis (1%) and visualized under UV light. The values on the left indicate the size of the bands. PM = molecular marker. **2B**: Undigested (lines:1-3-5-7-9-11) compared to digested with **BamHI** (lines:2-4-6-8-10-12) DNA; extracts were analyzed via agarose gel electrophoresis (1%) and visualized under UV light. The values on the left indicate the size of the bands. PM = molecular marker.

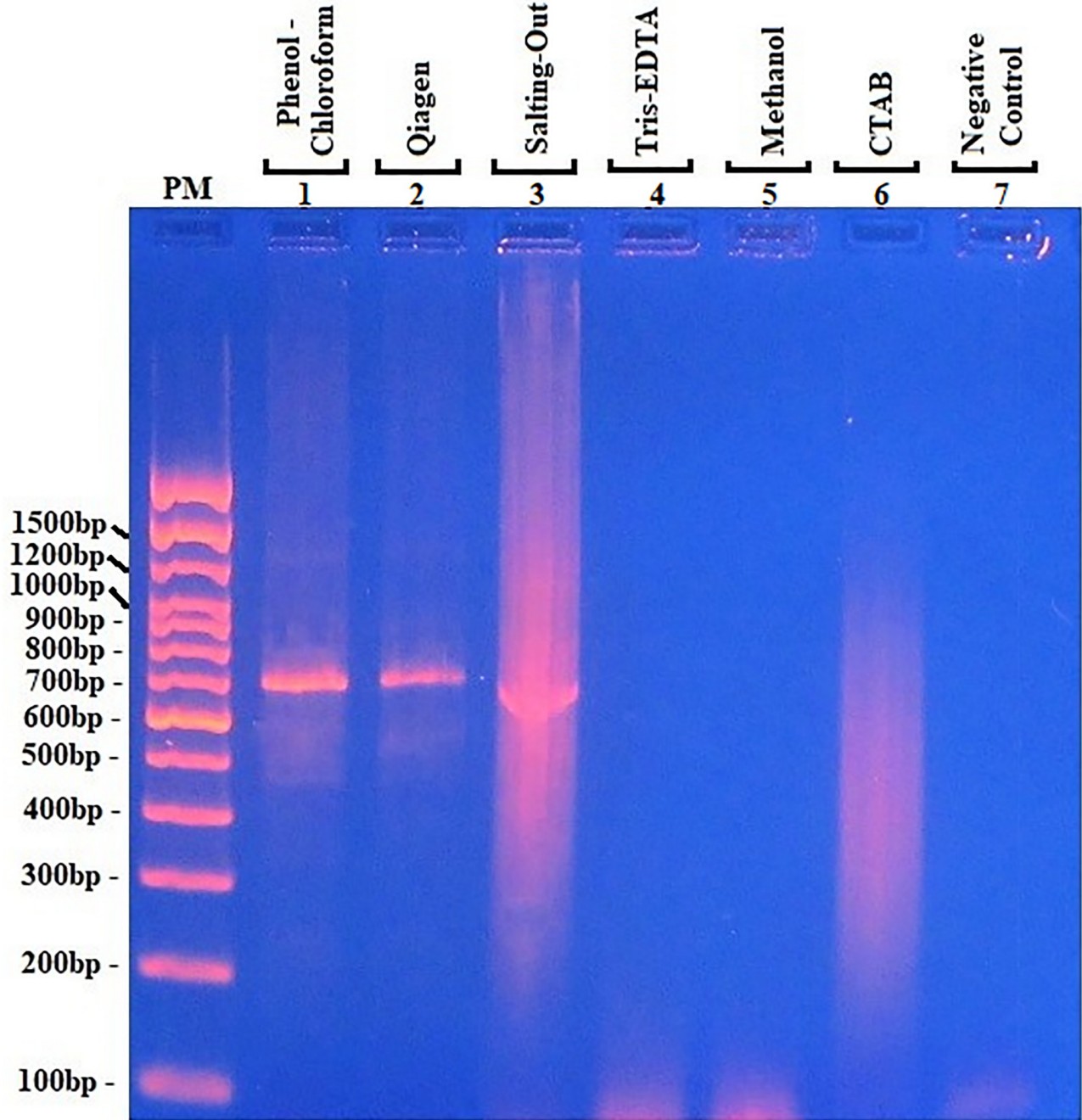

**Fig 3. Amplification of DNA extracted by PCR.** Comparison of amplicon obtained after amplification of *Loa loa* DNA extracted with the six methods as template. The primers were designed from the *Brugia ALT1* gene. Results of the analysis of amplicons after agarose gel electrophoresis (1.5%) and visualization under UV light. The methods are listed on the top of each band: phenol:chlorof line 1; Qiagen line 2; salting out line 3; Tris-EDTA line 4; Methanol line 5; CTAB line 6; band 7 is a negative control. PM = molecular marker. The values on the left represent the size of DNA.

methods (Qiagen, DNeasy® Blood & Tissue Kit). The integrity of the DNA and its purity were evaluated comparatively in terms of quality and quantity.

The DNA extracted with the phenol/chloroform and Qiagen methods was of good quality with slight contamination (Table 1). This was certainly due to the phenols or the salt,

according to the criteria defined in S1 Table. The yield when using phenol/chloroform was lower than that obtained with Qiagen (Table 1). The difference in the yield may be due to the repeated changing of the tube that is part of the phenol/chloroform process. The integrity and the concentration of DNA were evaluated by spectrometry and electrophoresis. Phenol/chloroform and Qiagen methods have been used with *L. loa* microfilariae [8, 17] and have also been used successfully with other parasites, i.e., *Leishmania* DNA from the urine of infected people [18]. The phenol/chloroform method has been reported to be sensitive and offers high yield and good-quality material [19]. However, the process is time-consuming and tedious because numerous steps are involved and the tubes have to be changed several times, which can result in a loss of DNA and can increase the risk of contamination. Another disadvantage is the high cost and increased hazard for the operator (i.e., phenol/chloroform and β-mercaptoethanol). All these problems make the method difficult to use in daily routine when many samples have to be tested as in the case of diagnostics. By contrast, the Qiagen method does not present any hazard but it is time-consuming and more expensive than the phenol/chloroform technique. The former method is difficult to institute as routine for laboratories in under-developed countries or in low-income centers. Both the phenol/chloroform and Qiagen methods are identically cuttable by endonuclease restriction enzymes and amplification by polymerase activity on the DNA, as evidenced by the PCR results (Fig 3, lines 1–2).

The yield obtained with the salting-out method was higher than that of phenol/chloroform and Qiagen (Table 1), and similar results were obtained in another study [20]. The purity of the DNA with salting-out was better than that from the rest of the methods, but the 260/230 ratio suggests some contamination by RNA (2.04) (Table 1). The electrophoretic profile showed a shift in the molecular size of extracted DNA after endonuclease EcoRI (Fig 2A) and BamHI (Fig 2B) treatment for extracts from phenol/chloroform, Qiagen, and salting-out, indicating the restriction activity on DNA extracted with these methods. Furthermore, amplicons were detected on DNA extracted with Qiagen, phenol/chloroform, and salting-out. The intensity of the amplicon band from salting out was similar to that generated with phenol/chloroform. This observation is in agreement with a previous study that showed the phenol/chloroform method was efficient for PCR assays [21]. However, the distortion observed with the amplicon obtained after PCR with the genomic DNA extracted by the salting-out amplicon is probably due to the sodium ion in the extract since sodium ions weaken the link between the sugar–phosphate of the DNA and the high concentration of sodium favors the solubility of DNA [22]. This study, like a previous study [23], suggests using an appropriate concentration of sodium ion to avoid extreme solubilization of DNA. Although the salting-out method is not commonly used, it is simple and cheap, there is no hazard involved, and it requires less processing time compared with phenol/chloroform and Qiagen, making it suitable for extraction of DNA from *L. loa* microfilariae.

The extraction methods using Tris-EDTA and methanol are simple and fast; for Tris-EDTA, the yield was low (0.552 μg) and the ratio suggested contamination (260/230 = 1.183) (Table 1). The electrophoretic profile after digestion by endonuclease activity (Fig 2A and 2B, lines 8) or PCR (Fig 3, line 4) did not show a band These results suggest either a small amount of DNA or the presence of contaminants; the fact that the DNA may be full of contaminants has been reported for *Plasmodium* [24]. The Tris-EDTA method could be adapted for field work as it does not involve changing the tubes, it is simple and cheap, and poses no hazard. However, there is a risk of false-negative results when using a high number of microfilariae, as used for this study.

For methanol, the yield was low (0.103 μg) and the ratios $A_{260}/A_{280} = 2.126$ and $A_{260}/A_{230} = 1.343$ suggest contamination probably because of the bad quality of DNA. There was no visible band before and after digestion with endonuclease activity or PCR. These results may be

explained by the fact that methanol did not destroy the sheath of the *L. loa* microfilariae. Another explanation could be that the extracted DNA was contaminated with methanol that probably acted as a polymerase inhibitor in the same way as ethanol and isopropanol [25].

The 260/230 and 260/280 ratios for CTAB were out of the normal limits for good quality DNA (2.426 and 2.01, respectively) (S1 Table) suggesting the DNA quality was not good. The results of electrophoresis on the 0.8% agarose gel revealed that the CTAB method generated a long smear without any specific band (Fig 1), and consequently endonuclease digestion (Fig 2A and 2B) and PCR amplification (Fig 3) also resulted in a smear. It should be noted that the CTAB method was set up for extraction of plant DNA and food of vegetable origin [26]. This method is especially convenient for elimination of polysaccharides and polyphenolic compounds not adapted for microfilaria DNA extraction. Furthermore, the duration of the process and the multiple changes of the sample vessel increase the possibility of contamination and destruction of the DNA integrity; besides, the cost of the reagents is very high.

In conclusion, the Tris-EDTA and methanol methods are simple, fast, cheap, and do not pose any hazard; however, in our study they were not sensitive for the extraction of *L. loa* microfilaria DNA. The methods using phenol/chloroform and the Qiagen kit are suited to the extraction of *L. loa* microfilaria DNA, but some reagents are expensive and present a hazard for the operator; they seem suitable for DNA recombinant technologies. The CTAB method is time-consuming and expensive, and it does not seem to be appropriate for *L. loa* DNA extraction. Salting-out is suitable for the extraction of DNA from *L. loa* microfilariae provided that the concentration of salt is adjusted during the process or eliminated before the end; it is also simple, cheap, and sensitive, without any inherent hazard. Moreover, it is fitting for recombinant DNA technology and for diagnostics based on the detection of DNA, as long as the correct concentration of sodium ion is used.

## Supporting information

**S1 Table. Formula for calculation of ratio, yield and interpretation.**
(DOCX)

**S2 Table. Raw replicate data (OD) and concentration of extracted DNA.**
(DOCX)

**S1 Raw images.** Fig 1. Comparison of six methods of *Loa loa* DNA extraction. PM = standard molecular weight. Lines 1–2: DNA from phenol/chloroform; 3–4: Qiagen extract; 5–6: salting-out extract; 7–8: Tris-EDTA extract; 9–10: methanol extract; 11–12: CTAB extract. The values on the left are the size of molecules. Fig 2. Digestion of extracted *Loa loa* DNA by EcoRI and BamHI endonuclease. **2A**: Undigested (lines: 1-3-5-7-9-11) compared to digested with **EcoRI** (lines: 2-4-6-8-10-12) DNA; PM = molecular marker. **2B**: Undigested (lines:1-3-5-7-9-11) compared to digested with **BamHI** (lines:2-4-6-8-10-12) DNA; extracts were analyzed via agarose gel electrophoresis (1%) and visualized under UV light. The values on the left indicate the size of the bands. PM = molecular marker. Fig 3. Amplification of DNA extracted by PCR. The methods are listed on the top of each band: phenol:chlorof line 1; Qiagen line 2; salting out line 3; Tris-EDTA line 4; Methanol line 5; CTAB line 6; band 7 is a negative control.
PM = molecular marker. The values on the left represent the size of DNA.
(PDF)

## Acknowledgments

This work was supported by the International Center for Medical Research, Franceville (CIRMF), Gabon.

## Author Contributions

**Conceptualization:** Roland Dieki, Elsa-Rush Eyang-Assengone, Edouard Nsi Emvo, Jean Paul Akue.

**Data curation:** Roland Dieki.

**Formal analysis:** Roland Dieki, Elsa-Rush Eyang-Assengone.

**Investigation:** Patrice Makouloutou-Nzassi, Félicien Bangueboussa.

**Methodology:** Edouard Nsi Emvo, Jean Paul Akue.

**Project administration:** Jean Paul Akue.

**Resources:** Patrice Makouloutou-Nzassi.

**Supervision:** Edouard Nsi Emvo, Jean Paul Akue.

**Validation:** Jean Paul Akue.

**Writing – original draft:** Roland Dieki, Elsa-Rush Eyang-Assengone.

**Writing – review & editing:** Edouard Nsi Emvo, Jean Paul Akue.

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
