## [Decision Letter · Decision Letter 0]

26 Nov 2021

PONE-D-21-31062Comparison of six methods for Loa loa genomic DNA extractionPLOS ONE

Dear Dr. Akue,

Thank you for submitting your manuscript to PLOS ONE. After careful consideration, we feel that it has merit but does not fully meet PLOS ONE’s publication criteria as it currently stands. Therefore, we invite you to submit a revised version of the manuscript that addresses the points raised during the review process.

Please see below the comments and suggested MINOR revisions made by the individual(s) who reviewed your manuscript.  If provided, the referee's report(s) indicate the revisions that need to be made before it can be accepted for publication.

We look forward to receiving your revised manuscript.

Kind regards,

Ricardo Santos

Academic Editor

PLOS ONE

Journal Requirements:

In your cover letter, please note whether your blot/gel image data are in Supporting Information or posted at a public data repository, provide the repository URL if relevant, and provide specific details as to which raw blot/gel images, if any, are not available. Email us at plosone@plos.org if you have any questions

Reviewers' comments:

Reviewer's Responses to Questions

**Comments to the Author**

1. Is the manuscript technically sound, and do the data support the conclusions?

Reviewer #1: No

Reviewer #2: Yes

2. Has the statistical analysis been performed appropriately and rigorously? 

Reviewer #1: N/A

Reviewer #2: N/A

3. Have the authors made all data underlying the findings in their manuscript fully available?

Reviewer #1: No

Reviewer #2: Yes

4. Is the manuscript presented in an intelligible fashion and written in standard English?

Reviewer #1: Yes

Reviewer #2: Yes

5. Review Comments to the Author

Reviewer #1: Abstract (perhaps include in the main manuscript next time for easier reviewing)

Why is DNA extraction linked with vaccine delivery here? (read further, change to development)

When you get down to the results section, it’s a lot of numbers and repeats of A260/A280. Perhaps think of a way to present without just having a wall of numbers.

Introduction

No need to put (L. loa) in brackets, just leave it at Loa loa and next mention L. loa.

Line 50: rather than name checking Qiagen here, I would just state that commercial DNA extraction kits exist. Obviously there are many of these from different companies, I wouldn’t put a specific kit here in the introduction. Qiagen is likely the most expensive kit – a consideration when it comes to thinking about wide spread use in low and middle income areas where parasites are often highly endemic.

Methods

Presumably the microfilariae came from human participants? Need to put in a little about how infected individuals were selected, consent forms and ethics, and blood collection.

Line 77: So there were 12 tubes in total each with 350,000 microfilariae in them? 2 for each technique?

Line 84: Put a reference in for Barker’s procedure

Line 86: Lysed how? Mechanical? Vortex? Heat?

Line 92: Stored dry or resuspended in buffer or water?

Line 95: Perhaps include a ref or link to the kit protocol – and state which protocol as most kits have a few different protocols in the manual depending on what material you have

Line 99: space between 5C°for

Line 100: space between number and units

Line 107: as per above procedure, stored dry or in

Line 112: reference for Miller

Tris-EDTA and methanol – these are quite crude extracts, any issues with inhibition in down stream applications (i.e. pcr)?

Line 166: Amplification of the extracted DNA? Either way, please ad in the PCR details (ah I see this is down further, perhaps switch the electrophoresis and PCR sections around). A real-time PCR would be interesting to check on efficiency and potential inhibition etc

Line 179/180: space between number and unit

Line 184: Where did the primers come from? Reference to paper they were first reported in. What is the target gene?

Discussion

Line 284: How was the contamination identified? What makes you certain it is salt and not something else foreign introduced into the prep?

Line 309: Indicating what activity?

It would be much easier to read the discussion if you could refer to tables or figures where the results are. Just stating yield was higher without the actual yield or reference to where the yield information is, isn’t very informative.

Line 311: Band intensity is not a great measure for DNA concentration. Particularly when doing it by eye.

Line 314: Distortion on the pcr amplicon, after endonuclease, just genomic DNA? Need to be clearer when talking about gel results as to which sample you are talking about.

Line 323: here you state the DNA concentration (by only one method, qubit?) for Tris-EDTA, and lower down for methanol. But it is not stated above for salting out, Qiagen or phenol/chloroform methods. Just a relative, higher or lower DNA. Refer to the table.

Line 338: Presumably for the PCR amplicon as just extracted genomic DNA would appear as a smear.

Line 352: Despite the band distortion? Based on Figure 3 I would not be recommending salting out unless some optimization/post DNA extraction step is taken to clean it up. I also wouldn’t recommended it on a time basis (table 1) as it didn’t work optimally, and it takes a long time to do. It’s only good point is that it is cheap. Normally you would say 2/3 – the three being time, cost, purity. Salting out is 1/3. At nest.

Was there any reason to use Qiagen as the commercial kit and not another, perhaps less expensive, commercial kit?

Figures: I know this is probably journal requirements, but I hate not having figure captions with the figures.

You should also be referring to these more often in the discussion – wherever you mention gel results, refer to the figure that shows it. Ditto the table.

2A and 2B are the replicates? Perhaps add that in the captions, Fig 2A replicate one for each extraction procedure pre and post endonuclease digestion etc.

Table: This appears to only be one of the replicates for each DNA extraction? Would like to see data for both replicates. Also is this Nanoview plus or Qubit results? Or both combined?

Reviewer #2: This is a relatively simple and straightforward report on comparing DNA extraction techniques for the diagnosis of an important human filarial nematode, Loa loa. Sensitive DNA extraction is key for diagnostics for this disease since many patients have low microfilaria concentrations, making diagnosis and appropriate treatment difficult. The authors compare six different extraction methods varying in cost and efficacy. Because cost and safety of use in the field is critical, this is an important contribution. The finding that the simple 'salting-out' method provides sufficient quality DNA for diagnostic tests and further evaluations is important and significant. Although not the most rapid method, it was highly efficient and could lead to more accurate diagnostics in the field.

I found this manuscript easy to follow for the most part. However, the Results section on Spectrometry Analysis was confusing and needs to be rewritten. In this section, the authors report a 260/280 ration for salting out as 1.9, yet in Table 1 they report 2.01, which is high and indicates contamination. The further analyses suggest the DNA extracted by 'salting-out' is of satisfactory quality for manipulation and diagnostics. There are other inconsistencies; they indicate on L197 that concentrations were 'far behind those of' when I think they mean 'far greater than.' Much of this text is redundant to Table 1, so perhaps the authors can combine the first two sections into one, entitled 'Comparative spectrometric analysis of the six methods'.

Overall, a very useful and practical contribution with the clarifications indicated above.

6. PLOS authors have the option to publish the peer review history of their article (what does this mean?). If published, this will include your full peer review and any attached files.

Reviewer #1: No

Reviewer #2: No

---

## [Author Response · Author response to Decision Letter 0]

9 Feb 2022

Response to Reviewers

A) Academic editor:

1. The manuscript meets the requirements of PLOS One as indicated in the template.

2. We have addressed the information on funding. We did not have a grant for this work but internal funding and we have changed the information accordingly.

3. The ethics statement is now included in the Methods section (see Methods, page 5, lines 104–105).

4. The images provided in this article are original, uncropped, and unadjusted. We added supporting information to clarify the analysis: (1) one part of the supporting information (S1) is related to data on the calculation and interpretation of the optical density (OD) ratio and yield; (2) the second part is related to the raw OD (S2) data from all the experiments and the mean of the replicat (X) from each of the six methods, including the mean concentration for each extraction method. 

5. The references have been completed by adding missing references in the original article (see page 5, line 109; page 7 line 131; page 8 line 160; page 11, line 233; page 11 line 234) and in reference section page 21, line 456-457; page 21, line 461-463, Line 464-466. 

The PLOS data policy: The rest of the complementary data underlying the findings described in the manuscript are now available in the supporting information Table S1 and Table S2.

B. Reviewer 1: 

Abstract:

An abstract has been included in the main manuscript (pages 2 and 3, line 23–56)

DNA extraction: The term “vaccine delivery” has been replaced by “development” (see line 25, page 2).

Results: The presentation of the results has been shortened to avoid repetition (see page 2, line 40–42).

Introduction:

Loa loa: The term has been written out, instead of L. loa, the first time it is used (see page 4, line 65).

Line 50: The sentence has been changed in accordance with the reviewer’s comments (see line 91, page 5).

Methods:

Human participants: A sentence has been added to address this point (see page 5, line 104–109),

Line 77: Since the meaning of the sentence seemed to be ambiguous, it has been rewritten (see page 6, line 124–125).

Line 84: The reference has been added (see page 7, line 131) and included in the reference section (see page 21, line 456).

Line 86: This has been corrected (see page 7, line 133–134).

Line 92: This is now specified (see page 7, line 139-140). 

Line 95: The protocol is now mentioned (see page 7, line 143-144). 

Line 99: There is now a space between temperature and “for” (see page 7, line 148).

Line 100: There is now a space between numbers and units (see page 7, line 148).

Line 107: The storage conditions are now specified (see page 8, line 156).

Line 112: The reference was added in the text (see page 8, line 160) and in the reference section; page 21, line 457-458. 

Line 166: Because of confusion in the original form, we have changed the presentation and switched the PCR and electrophoresis section around as recommended by the reviewer (page 11, line 224–242).

Line 179/180: A space has now been inserted between all numbers and units (see page 11, line 227-228)

Line 184: The origin of primers, the reference paper, and the target gene are now specified (page 11, line 233-234).

Discussion:

Line 284: Identification of the contaminant was carried out using the classic parameters described in supporting information Table S1 backed by raw data in supporting information Table S2. Page 15, line 319-321 and in section material and method page 10, line 214-215.

Line 309: The activity is now specified in line 356-358, page 17

As suggested by the reviewer, we now refer to the figures and table in the relevant parts of the discussion section. 

Line 311: To avoid any bias that may arise from the part on “band intensity by eye,” the sentence was removed from the original text (see page 19, line 403–405).

Line 314: The statement on “band distortion” has been clarified (page 17, line 362–366).

Line 323: Table 1 is referred to systematically throughout the discussion section wherever the concentration or yield is mentioned.

Line 338: This detail has been added (see page 15, line 307; page 18, line 390).

Line 352: The sentence has been modified according to the reviewer’s comments (see page 19, line 403–405).

-The Qiagen kit was chosen on the basis of availability and because this kit is typically used in our laboratory.

- As the reviewer may know, figures were not included in the text because this is a requirement of the journal. WE think that the reviewer and editor will understand why. Furthermore, we now refer to the figure and the table wherever possible.

C. Reviewer 2:

The reviewer notes several inconsistencies such as in the reporting of the 260/280 ratio. This has been corrected (page 12, line 247-248).

In line 197, “far behind those of” has been changed to “far greater than” (see page 12, line 254). Also, to address some other inconsistencies and redundancies, the whole section has been rewritten and is now entitled, as suggested by the reviewer, “Comparative spectrometric analysis of the six methods” (page 12, line 245–264.)

---

## [Decision Letter · Decision Letter 1]

7 Mar 2022

Comparison of six methods for Loa loa genomic DNA extraction

PONE-D-21-31062R1

Dear Dr. Akue,

We’re pleased to inform you that your manuscript has been judged scientifically suitable for publication and will be formally accepted for publication once it meets all outstanding technical requirements.

Kind regards,

Ricardo Santos

Academic Editor

PLOS ONE

Additional Editor Comments (optional):

Reviewers' comments:

Reviewer's Responses to Questions

**Comments to the Author**

1. If the authors have adequately addressed your comments raised in a previous round of review and you feel that this manuscript is now acceptable for publication, you may indicate that here to bypass the “Comments to the Author” section, enter your conflict of interest statement in the “Confidential to Editor” section, and submit your "Accept" recommendation.

Reviewer #1: All comments have been addressed

Reviewer #2: All comments have been addressed

2. Is the manuscript technically sound, and do the data support the conclusions?

Reviewer #1: (No Response)

Reviewer #2: Yes

3. Has the statistical analysis been performed appropriately and rigorously? 

Reviewer #1: (No Response)

Reviewer #2: Yes

4. Have the authors made all data underlying the findings in their manuscript fully available?

Reviewer #1: (No Response)

Reviewer #2: Yes

5. Is the manuscript presented in an intelligible fashion and written in standard English?

Reviewer #1: (No Response)

Reviewer #2: Yes

6. Review Comments to the Author

Reviewer #1: (No Response)

Reviewer #2: (No Response)

7. PLOS authors have the option to publish the peer review history of their article (what does this mean?). If published, this will include your full peer review and any attached files.

Reviewer #1: No

Reviewer #2: No

---

## [Editor Report · Acceptance letter]

11 Mar 2022

PONE-D-21-31062R1 

Comparison of six methods for *Loa loa* genomic DNA extraction 

Dear Dr. Akue:

I'm pleased to inform you that your manuscript has been deemed suitable for publication in PLOS ONE. Congratulations! Your manuscript is now with our production department. 

Kind regards, 

on behalf of

Dr. Ricardo Santos 

Academic Editor

PLOS ONE